# Enzymatic properties of a L-asparaginase without secondary glutaminase activity from *Streptomyces scabrisporus*

Ricardo Rodríguez-Vargas[1], Francisca Villanueva-Flores[2],
María Fernanda Gutiérrez-Chávez[1], Carlos Medrano-Villagómez[1],
Andrés Zárate-Romero[1], Alejandro Huerta-Saquero[1]*

1 Centro de Nanociencias y Nanotecnología, Universidad Nacional Autónoma de México, Ensenada, Baja California, México, 2 Centro de Investigación en Ciencia Aplicada y Tecnología Avanzada Unidad Morelos del Instituto Politécnico Nacional, Boulevard de la Tecnología, Xochitepec, Morelos, México

* saquero@ens.cnyn.unam.mx

## Abstract

Acute lymphocytic leukemia (ALL) is characterized by the uncontrolled proliferation of lymphocyte precursor cells within the bone marrow, blood and extramedullary sites. L-asparaginase has become a standard treatment in childhood cases of ALL by reducing the asparagine levels in the bloodstream on which leukemic cells depend, as they cannot synthesize it. The reduction of asparagine leads to cell cycle arrest and death by apoptosis. However, due to the bacterial origin of L-asparaginase, it causes immunogenic reactions, and the cross-glutaminase activity that the enzyme exhibits cause ammonium accumulation and toxicity in different organs and tissues. Enzymes with a lower immunogenic profile that preserve their affinity for the substrate asparagine and that do not have glutaminase activity are needed, such as L-asparaginases from *Streptomyces scabrisporus* or *Rhizobium etli*. In this work, the L-asparaginases from *S. scabrisporus* and *R. etli* were purified and characterized, and the kinetic parameters of the enzymes were compared under physiological conditions. Furthermore, both enzymes reduced the viability of MOLT-4 leukemic cells in a time- and concentration-dependent manner.

## Introduction

Acute lymphocytic leukemia (ALL) is the most prevalent type of cancer in children in developed countries around the world. The study of ALL has allowed the classification of this disease according to the cell lineage by which it originated (B or T cells), the state of cellular differentiation, or by different groups of loci that are precursors of this cancer [1]. One of the main characteristics present in ALL is the monoclonal or oligoclonal proliferation of hematopoietic precursor cells within the bone marrow [2]. The clinical symptoms include fever and enlargement of the liver,

**Data availability statement:** All relevant data are within the manuscript.

**Funding:** The author(s) received no specific funding for this work.

**Competing interests:** The authors have declared that no competing interests exist.

spleen, and lymph nodes. Among the disorders associated with lymphoid tissue are anemia, thrombocytopenia, and granulocytopenia [3]. L-asparaginase's mechanism of action in treating ALL is currently widely described; the enzyme hydrolyzes the amino acid asparagine, converting it into aspartic acid and ammonium [4]. This allows reducing the levels of asparagine found in the bloodstream. Because leukemic cells cannot synthesize asparagine, they enter a state of prolonged nutritional stress that leads to proliferative arrest and, subsequently, to apoptosis of the leukemic cells [5]. The use of L-asparaginase of bacterial origin, has become a standard treatment in cases of childhood ALL; however, in the case of adults with this condition, the use of L-asparaginase has an increase in mortality caused by toxic effects and immunogenic reactions due to the bacterial origin (*Escherichia coli* or *Erwinia chrysantemi*) of this enzyme, as well as its glutaminase cross-activity [6,7]. Glutaminase activity contributes to adverse effects, including acute pancreatitis, thrombosis, hyperglycemia, leukopenia, and central nervous system hemorrhages [8–10]. Searching for alternative L-asparaginases for the ALL treatment, those with lower immunogenicity than *E. coli* L-asparaginase, and no glutaminase activity, are required.

In this sense, asparaginase activity has been observed in a variety of microorganisms, including fungi, bacteria and yeasts. However, this activity is closely associated with glutaminase activity in most organisms examined [11]. Numerous known actinobacteria currently produce various drugs, antibiotics, or compounds of medical interest. Among these compounds is L-asparaginase, produced by different species of the *Streptomyces* genus [12,13]. In our previous work, we characterized *Rhizobium etli* asparaginase, an enzyme that did not show glutaminase activity [14,15]. On the other hand, searching for homologous sequences to type II asparaginases from *Escherichia coli* and *Streptomyces coelicolor* in actinobacteria species resulted in the discovery of an enzyme with possible asparagine binding sites, which theoretically does not use glutamine as a substrate and has a low immunogenic profile in comparison with *E. coli* asparaginase enzyme [16]. Immunoinformatics studies allowed us to predict the antigenicity as well as the number of epitopes of this enzyme. *S. scabrisporus* L-asparaginase has a low number of T cell epitopes, a low antigenicity profile, and low allele coverage in the population compared to those from *E. coli* L-asparaginase. These values are closely linked to the population that develops an immune response to the enzyme. The *S. scabrisporus* enzyme belongs to the PF06089.11 protein family, in which *Rhizobium etli* L-asparaginase can also be found [16]. This enzyme was reported as a metalloprotein with an affinity for zinc that plays a fundamental role in the organization of residues near the enzyme's active site [17,18].

Using new proteins with lower immunogenicity profiles and asparaginase activity levels similar to those of current commercial enzymes, without cross-glutaminase activity, represents an alternative for treating ALL. In this work, the recombinant enzyme L-asparaginase obtained from *S. scabrisporus* was biochemically characterized. It was expressed in *E. coli*, purified, and its kinetic profile was compared to *R. etli* L-asparaginase under physiological conditions. MOLT-4 leukemic cell viability assays revealed that both enzymes induce cell death in a time- and

concentration-dependent manner. For all the above, these enzymes are proposed as a chemotherapeutic agent without glutaminase activity for their potential use in treating ALL.

## Results

### *S. scabrisporus* and *R. etli* asparaginases purification

The L-asparaginases from *S. scabrisporus* and *R. etli* were purified to near homogeneity by affinity chromatography using a HisTrap column (Cytiva). Electrophoretic analysis of the purified L-asparaginase from *S. scabrisporus* showed a 34 kDa band, consistent with the molecular weight of the recombinant enzyme. A protein yield of 2 mg/L was obtained. On the other hand, the purified L-asparaginase from *R. etli* showed a 38 KDa band, according to the estimated molecular weight of the recombinant enzyme. A protein yield of 3 mg/L was obtained (Fig 1) [17].

### pH optimum, thermal stability and glutaminase cross-activity of *S. scabrisporus* L-asparaginase

The optimal pH for enzymatic activity was determined by 30-minute kinetics assays using different buffers (acetate pH 3–5, sodium phosphate pH 6–8, carbonate 9–11). The highest enzymatic activity was observed at pH 10 (Fig 2A). No significant activity differences were detected in the rest of the conditions evaluated. The optimal pH values are similar to those reported for other asparaginases of the same family, such as that of *R. etli* [17]. To determine the enzyme's kinetic parameters, enzymatic kinetics assays were performed with reaction times of 0–10 minutes to determine the initial reaction rate and approximate the Michaelis-Menten constant, at optimum pH of 10. Substrate (asparagine) concentrations ranging from 0.5 to 20 mM, were evaluated. A Michaelis-Menten constant (Km) of 7.361 mM was determined, which presented a maximum reaction rate (Vmax) of 6.08 µmol/s$^{-1}$ while the catalysis constant (kcat) was 40.832 s$^{-1}$ (Fig 2B).

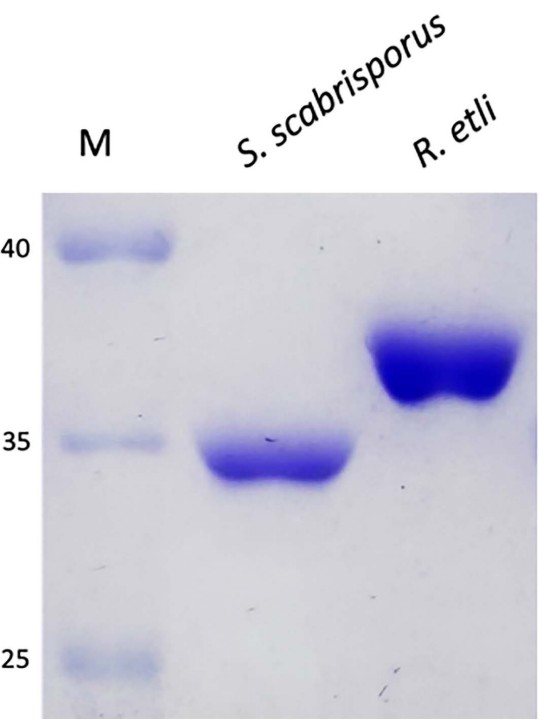

**Fig 1. Purified recombinant L-asparaginases from *S. scabrisporus* and *R. etli*.** SDS-PAGE at 12%: **M.**- Molecular weight markers (in kDa); lane 1) *S. scabrisporus* L-asparaginase (34 kDa); lane 2) *R. etli* L-asparaginase (38 kDa). Protein bands were stained with Coomasie Brilliant Blue R-250.

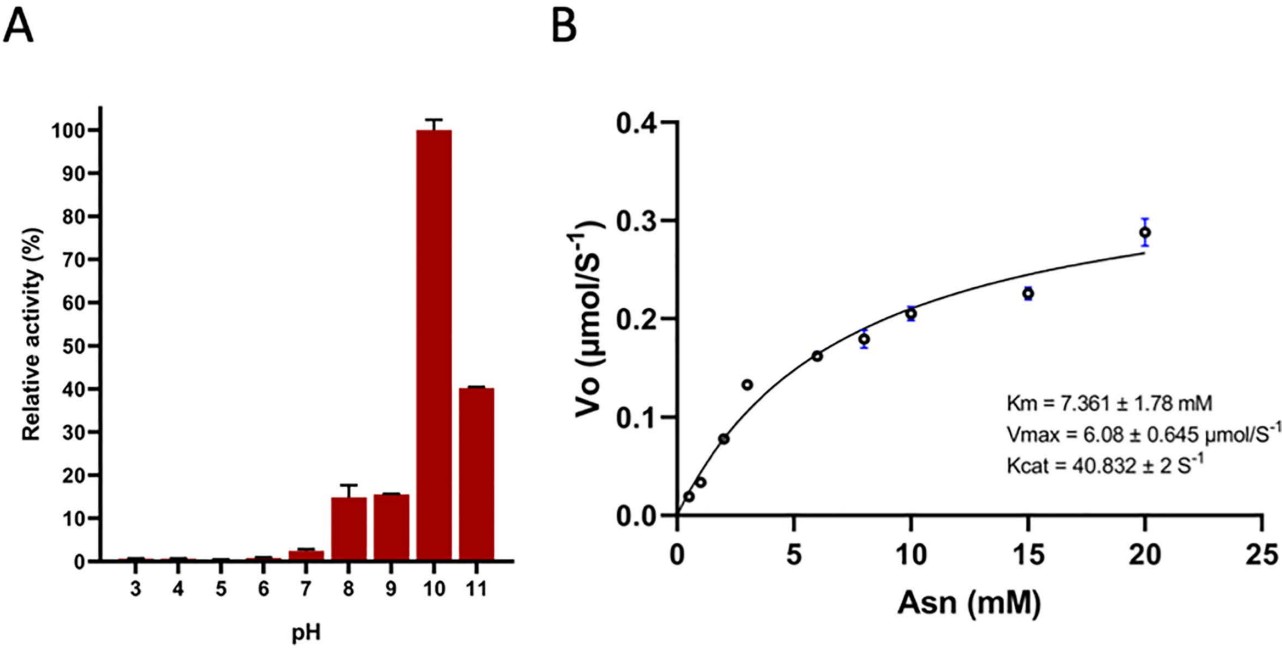

**Fig 2. pH optimum and kinetic parameters of *S. scabrisporus* asparaginase.** Enzymatic activity was determined at different pH (3 to –11) (A) Optimum pH was determined as 10. (B) At this pH, kinetic parameters were calculated and adjusted to the Michaelis-Menten model, shown in the inset.

To evaluate the stability of the enzyme, enzymatic activity was measured after incubating the enzyme at 37 °C for 17, 24, and 48 hours and after storing the enzyme for 6 weeks at 4 °C (Fig 3). The enzymatic activity for the different conditions evaluated is affected by storage conditions.

The comparison of the enzymatic activity of purified fresh asparaginase showed that storing the enzyme for 6 weeks at 4 °C presents a 60% reduction in the relative activity of the enzyme compared to the enzyme without incubation. On the other hand, the enzyme incubated at 37 °C for 16 and 24 hours presents a 22% reduction in activity, while incubation for 48 hours led to a 59% reduction (Fig 3).

The glutaminase activity of the *S. scabrisporus* asparaginase enzyme was then evaluated. The secondary glutaminase activity (Fig 4) was measured using enzymatic kinetics over a 60-minute time course using individual reactions containing 10 mM of either asparagine or glutamine as a substrate. The results showed that *S. scabrisporus* asparaginase does not show secondary glutaminase activity.

### Kinetic parameters comparison of L-asparaginases from *S. scabrisporus* and *R. etli* under physiological conditions

For the comparison of the kinetic parameters of both enzymes, enzymatic kinetics were carried out with incubation times of 0–10 minutes, and the results were fitted to the Michaelis-Menten model. Physiological pH and temperature conditions were used to simulate conditions in the human body (pH 7 and 37 ºC). Different substrate concentrations of asparagine were tested, ranging from 0.5 to 20 mM and 0.5 to 60 mM. In the case of the purified enzyme from *S. scabrisporus* (Fig 5A), it was found that the activity of the enzyme was reduced when the pH was changed, which can be reflected in the change in kinetic parameters such as Km, which under optimal pH conditions was $7.361 \pm 1.78$ mM and under physiological conditions increased approximately 8 times resulting in a Km of $60.49 \pm 10.53$ mM indicating a decreased substrate affinity. On the other hand, the kinetics performed with the *R. etli* enzyme under physiological conditions (Fig 5B) also

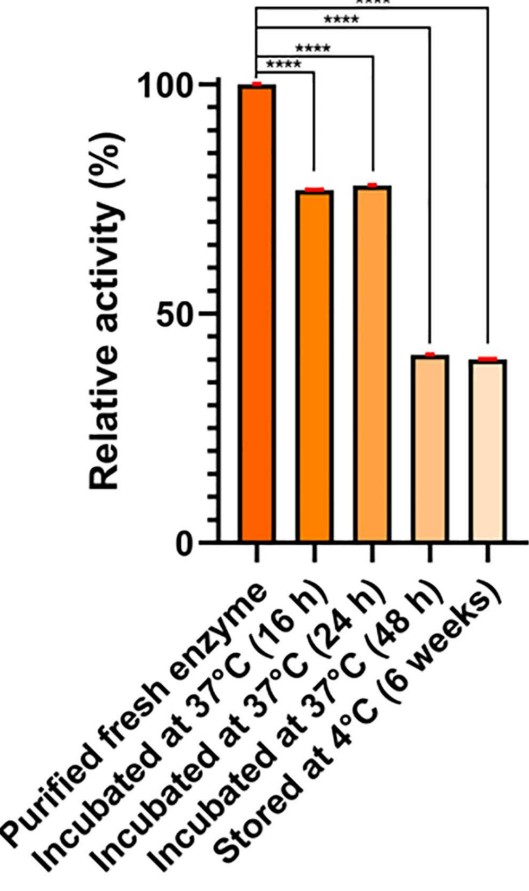

**Fig 3. Stability of *S. scabrisporus* asparaginase.** Enzymatic activity was measured with purified fresh enzyme, or after incubated the enzyme at 37 ºC for 16 h, 24 h, or 48 **h.** The activity of stored enzyme at 4 ºC for 6 weeks is also shown. Statistical analysis was performed using a two-way ANOVA with a Dunnett´s post hoc test and α = 0.05. ****Denotes significative differences between treatments and condition control.

showed a 3-fold increase in Km under physiological conditions compared to its optimal activity pH, which had already been previously reported (1).

### *S. scabrisporus* and *R. etli* asparaginases reduce cell viability of MOLT-4 cells

To determine whether asparaginases from *S. scabrisporus* and *R. etli* maintained their activity under cell culture conditions and were able to alter the viability of leukemic cells, we performed cell proliferation assays with MOLT-4 leukemic cells exposed to different concentrations of asparaginase, ranging from 1.56 to 50 U/mL (Fig 6).

Cell viability was quantified using the MTS assay after 24, 48, and 72 hours of incubation in the presence of *R. etli* and *S. scabrisporus* asparaginases and Leunase (*E. coli* L-asparaginase) as a reference control. Commercial *E. coli* asparaginase (Leunase) showed the highest cytotoxicity, reducing cell viability by 50% even at the lowest enzyme concentration (1.56 U/mL) after 24 hours of incubation (Fig 6A) and eliminating almost 100% of leukemic cells after 48 and 72 hours of incubation (Fig 6B and 6C).

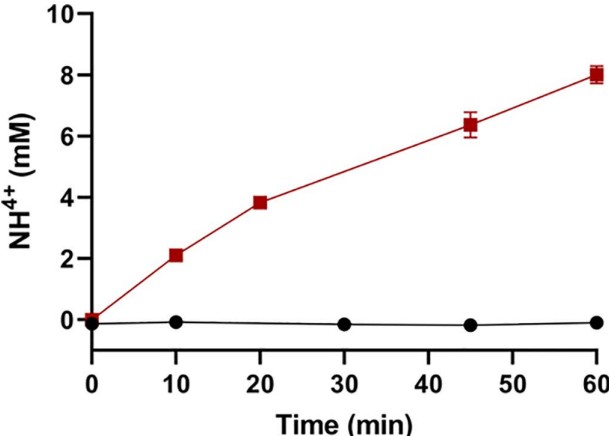

**Fig 4. S. scabrisporus asparaginase showed no glutaminase activity.** The graph shows ammonium released by enzymatic activity using asparagine (red) or glutamine (black) as a substrate.

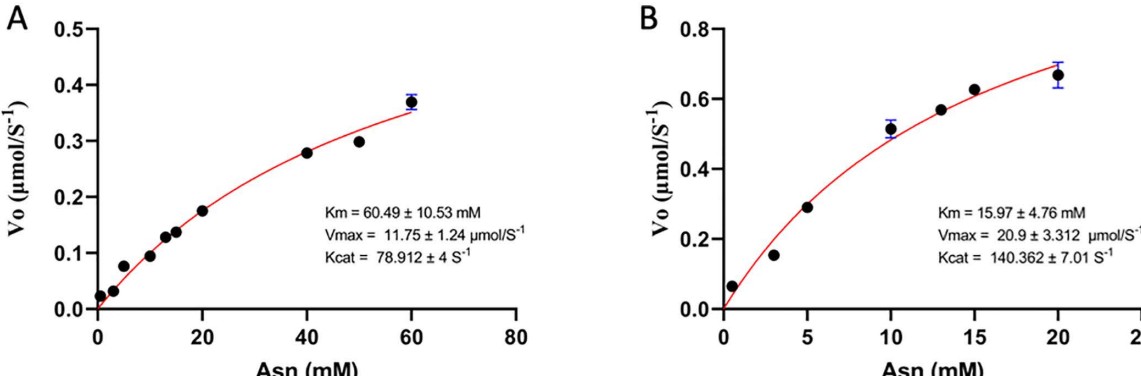

**Fig 5. Enzymatic activity of asparaginases. (A)** *S. scabrisporus* asparaginase and **(B)** *R. etli* asparaginase. Initial activities were adjusted to the Michaelis-Menten model. Kinetic parameters are shown in the insets.

*R. etli* asparaginase also showed activity, reducing cell viability by 50% during the first 24 hours of incubation. However, unlike Leunase, this reduction was observed with 50 U/mL (Fig 6A). After 48 and 72 hours, *R. etli* asparaginase decreased cell viability partially and dose-dependently but without reaching the levels achieved with Leunase (Fig 6B and 6C).

*S. scabrisporus* asparaginase showed the least inhibition of cell viability during the first 24 hours, reaching a viability reduction of 35% at the highest enzyme concentrations (Fig 6A). However, during the 48 and 72 hours of incubation, it showed consistent activity, reducing cell viability by up to 80% at enzyme concentrations of 6.25, 12.5, 25 and 50 U/mL (Fig 6B and 6C).

## Discussion

The search for new sources of L-asparaginase is an imperative task to address the adverse effects caused by the secondary activity of glutaminase, which can lead to hyperammonemia due to the excessive accumulation of ammonia in the body from glutamine hydrolysis [19]. This problem has been reported in asparaginases from *E. coli* and *E. chrysanthemi*.

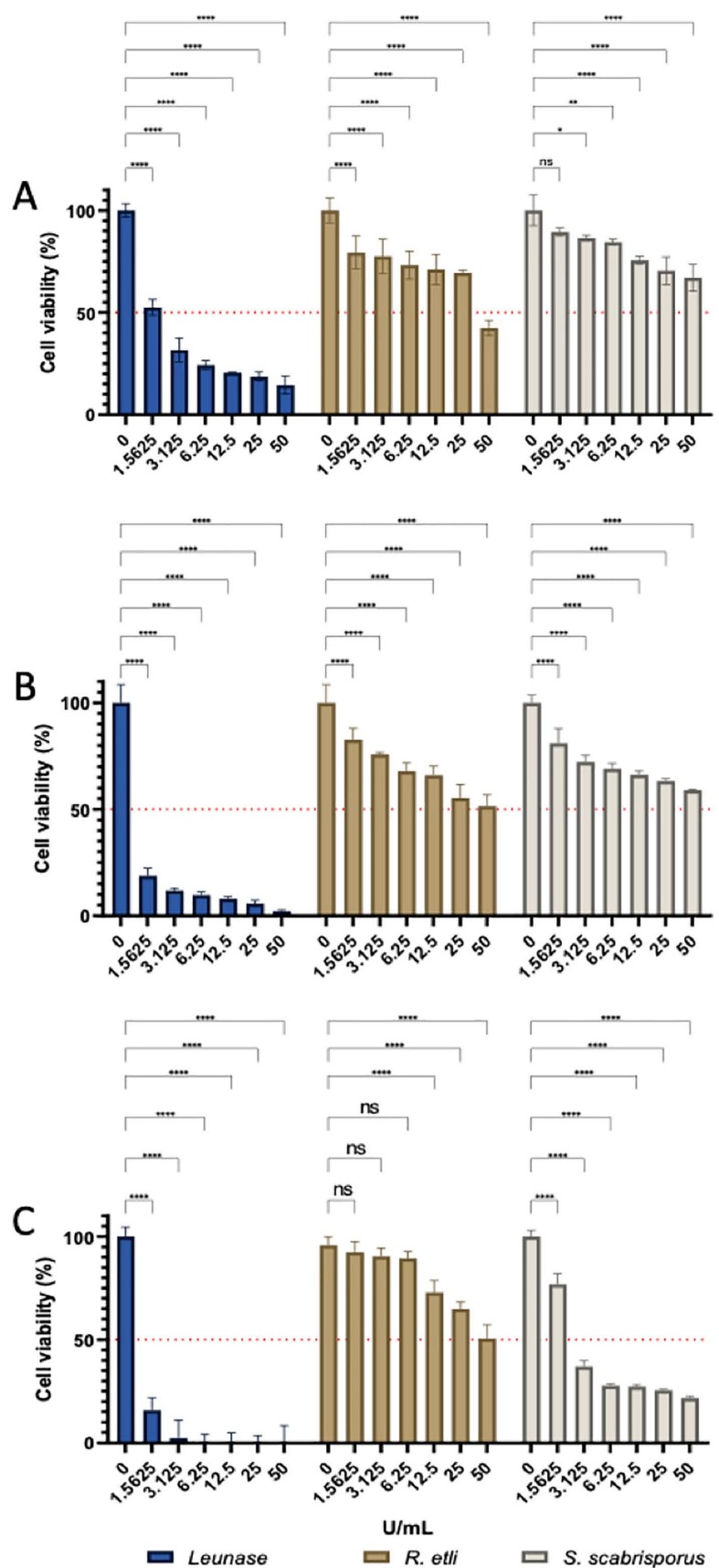

U/mL

| ■ *Leunase* | ■ *R. etli* | □ *S. scabrisporus* |

**Fig 6. Cell viability assays.** MOLT-4 leukemic cells were incubated with Leunase (blue), *R. etli* asparaginase (brown) and *S. scabrisporus* asparaginase (gray) during A) 24, B) 48, and C) 72 hours. Red dotted lines indicate 50% of cell viability. Cell viability was quantified by MTS assay by triplicate. Statistical analysis was performed using a two-way ANOVA with a Dunnett´s post hoc test and $\alpha = 0.05$. Significance values $* < 10^{-1}$, $** < 10^{-2}$, $*** < 10^{-3}$, $**** < 10^{-4}$.

Another problem associated with asparaginases currently used as treatment includes the effects caused by immunogenic response ranging from a mild allergic reaction to anaphylactic shock [20]. Therefore, finding L-asparaginase variants with high substrate specificity and zero secondary glutaminase activity is a fundamental objective. Such variants would improve treatment safety and efficacy by reducing the required doses and associated adverse effects [20]. Recent research has identified potential new sources with these characteristics, such as L-asparaginase from *R. etli* [14] and the one described in this work obtained from *S. scabrisporus*.

We focused our efforts on amplifying the gene encoding L-asparaginase from *S. scabrisporus*, cloning and expression. Afterwards, the purification protocol resulted in obtaining high-purity L-asparaginase from *S. scabrisporus*. The yield of soluble protein obtained was 2 mg/L, sufficient to perform the enzyme kinetics assays and biochemical characterization.

The enzyme characterization process included evaluating activity at 37 °C, taking human body temperature as a reference since the enzyme is intended as a potential chemotherapeutic agent. In determining the optimal pH at which the enzyme hydrolyzed the amino acid asparagine, it was found that the ideal pH for the enzyme is in the alkaline range, with an optimum at pH 10. The result obtained for the optimal pH for this L-asparaginase represents a limitation for its potential therapeutic use since the physiological pH at which this enzyme is required is within a range of 7.35–7.45. However, different approaches have been reported to address this problem. The immobilization of free L-asparaginase on magnetic ferric oxide ($Fe_3O_4$) nanoparticles can improve the enzyme's stability over a wider pH range. However, it is important to note that other kinetic parameters, such as Km and Vmax, may be altered compared to the values obtained under optimal conditions [21–23].

The kinetic parameters observed for *S. scabrisporus* L-asparaginase showed that the enzyme has a Km of 7.361 mM, a Vmax of 6.08 µmol/s, and a kcat of 40.832 s-1 under optimal pH conditions. A comparison of the kinetic parameters with other currently characterized asparaginases is presented in Table 1.

An important feature to highlight from the kinetic characterization of the enzyme is that this work experimentally corroborated the findings of the bioinformatics analysis conducted by González-Torres et al. (2020) regarding the lack of secondary glutaminase activity in *S. scabrisporus* L-asparaginase (Fig 4), which represents an advantage over currently used therapeutics asparaginases.

The enzymatic stability is critical in the field of clinical application, as current L-asparaginase formulations often require the addition of polyethylene glycol (PEG) to improve the stability of the enzyme and extend its half-life [25]. In the case of L-asparaginase from *E. chrysanthemi*, once reconstituted from its lyophilized form, it has a shelf life of 8 hours, while when stored at temperatures of 4 °C, its shelf life can be extended by up to 120 hours while retaining 98% of its initial activity [26]. In the case of *E. coli*, it has been reported that under non-optimized storage conditions at a temperature

**Table 1. Kinetic parameters of L-asparaginases.**

| Source | *Km* (mM) | *kcat* (s⁻¹) | Glutaminase activity | References |
|---|---|---|---|---|
| *E. coli* | $0.02 \pm 0.03$ | 440 | Yes | [24] |
| *E. chrysanthemi* | 0.05 | $207 \pm 358$ | Yes | [24] |
| *R. etli* | $4.2 \pm 0.03$ | $438 \pm 32$ | No | [17] |
| *S. scabrisporus* | $7.361 \pm 1.78$ | $40.832 \pm 2$ | No | (This work) |

between 4–8 °C, the enzyme retains 92% of its activity after 168 hours [27]. These results contrast with those observed for *S. scabrisporus* asparaginase, which preserves about 40% of its activity after six weeks of storage at 4 °C. This is an important factor to consider since the regimen of these enzymes requires multiple applications in patients to reduce blood asparagine levels. Currently, there are no reports on the stability of *R. etli* asparaginase under storage conditions at 4 °C or after prolonged incubation at 37 °C.

Once the asparaginases from *R. etli* and *S. scabrisporus* were purified, the kinetics of both enzymes were evaluated under physiological pH and temperature conditions. In the case of *R. etli* asparaginase, a Vmax of 20.9 ± 3.3 µmol/s, Km of 15.9 ± 4.7 mM, and a $K$cat of 140.3 ± 7 s-1 were obtained. These parameters are consistent with those previously described by Loch et al. (2021). In the case of *S. scabrisporus* asparaginase, changes were observed in the kinetic parameters, particularly a significant increase in Km, which rose approximately eightfold to 60.5 ± 10.5 mM. This result was expected because the optimal pH of the enzyme is in an alkaline range. However, one of the interesting changes observed was an increase in $K$cat, which doubled to a value of 78.9 ± 4 s-1. These changes in enzyme kinetics are unusual and have been previously attributed to mutations in the enzyme sequence [28].

Despite the non-competitive kinetic parameters of *S. scabrisporus* asparaginase compared to those of the asparaginases used in the current treatment at physiological pH, the aforementioned characteristics of null glutaminase activity and its stability during storage motivated us to determine its antiproliferative activity on MOLT-4 leukemic cells. To our surprise, *S. scabrisporus* asparaginase reduced the viability of the cell line in a concentration- and time-dependent manner, although to a lesser extent than *E. coli* (Leunase) and *R. etli* asparaginases, during the first 24 hours of treatment (Fig 6A). After 48 hours of incubation, *S. scabrisporus* asparaginase activity was comparable to that of *R. etli*, both being less effective than that of *E. coli* (Fig 6B). However, at 72 hours, *S. scabrisporus* asparaginase activity significantly reduced cell viability, outperforming *R. etli* asparaginase and reaching highly competitive levels of inhibition from a concentration of 6.25 Units/mL, reducing MOLT-4 cell viability by 75% (Fig 6C). The high stability of *S. scabrisporus* asparaginase can explain this result, suggesting that, despite its lower catalytic capacity, it persists for a longer time, reducing the availability of asparagine in the medium and, consequently, reducing cell viability at longer incubation times.

These results support the notion of the stability of this enzyme and its efficient glutaminase-free asparaginase activity, highlighting its potential as a viable alternative for the treatment of ALL in cases where patients are hypersensitive to ammonium production or have already generated an immune response to currently used asparaginases. It also enables the protein engineering of this asparaginase to enhance its activity at physiological pH through site-directed mutagenesis. The field is open for the development of biobetters for the treatment of ALL [29].

## Conclusions

The L-asparaginases from *S. scabrisporus* and *R. etli* were purified and characterized; both enzymes reduced the viability of MOLT-4 leukemic cells in a time- and concentration-dependent manner. Despite the low affinity of *S. scabrisporus* asparaginase for its substrate, asparagine, under physiological conditions, its stability and efficient glutaminase-free asparaginase activity highlight its potential as a viable alternative for the treatment of ALL.

## Materials and methods

### Cloning of the genes encoding L-asparaginases and expression of the recombinant protein from *S. scabrisporus* and *R. etli* in *E. coli*

The gene encoding L-asparaginase from *S. scabrisporus* (ENA, OPC79493.1) was synthesized with codon usage optimization for expression in *E. coli*. The gene was cloned using the pJET vector (Thermo Fisher) with blunt ends and cut with the restriction enzymes EcoRI and NdeI and then subcloned into the plasmid pET28a (+) to obtain the pET28a-AnsSs construct. The gene encoding type II L-asparaginase from *R. etli* (Q2K0Z2) was cloned in the same way, to which the restriction sites NcoI at the 5' end and EcoRI at the 3' end, respectively, were added. Subsequently, the plasmid pET28a-AnsRe was

obtained, with which BL21 (DE3) cells were transformed to obtain the clones BL21/pET28a-AnsRe. The transformation process was performed by the heat shock technique using chemically competent BL21 (DE3) *E. coli* cells. The transformant strains BL21/pET28a-AnsSs and BL21/pET28a-AnsRe were plated on an LB agar plate supplemented with kanamycin (35 µg/mL), and positive colonies were selected.

For the pre-inoculum of the cultures, 200 ml of LB medium supplemented with 35 µg/mL kanamycin was used and incubated for 20 h at 37 ºC with constant shaking at 180 rpm. From this cultures, 2 L of LB medium + 35 µg/mL kanamycin, 1 mM zinc chloride ($ZnCl_2$) was inoculated separately and incubated at 37ºC with constant shaking at 180 rpm. Optical density was monitored at 600 nm using the Thermo Scientific™ Multiskan™ GO Microplate Spectrophotometer, and once an optical density of 0.5–0.7 was reached, the cultures were induced with 200 µM IPTG (Isopropyl β-D-1-thiogalactopyranoside). The culture was incubated at 30 ºC and 180 rpm for 20 h. Induced cultures were harvested by centrifugation and cell pellets resuspended in 60 mL of lysis-binding buffer (50 mM $Na_2HPO_4$-$NaH_2PO_4$ pH 7.4, 150 mM NaCl, 2 mM ß-mercaptoethanol and 10% glycerol) with 100 µL of lysozyme (20 µg/µL) and 300 µl of PMSF (200 mM), incubated for 1 hour at 37 °C and lysed by sonication for 15 min in pulses of 10 seconds with 5 seconds of rest. The sonicated suspension was centrifuged, and the supernatant was injected into a HisTrap column (Cytiva) previously equilibrated with a lysis-binding buffer. Subsequently, the column was subjected to a washing step with 50 mL of 20 mM imidazole. Bound proteins were eluted with elution buffer (50 mM $Na_2HPO_4$-$NaH_2PO_4$ pH 7.4, 150 mM NaCl, 2 mM ß-mercaptoethanol, and 10% glycerol) using a gradient of imidazole concentrations of 100, 300, and 500 mM. The presence of the enzymes was analyzed in all fractions by electrophoresis in a 12% denaturing polyacrylamide gel. Subsequently, dialysis of the imidazole fractions was performed using a 12–14 kDa dialysis membrane in 1 L of dialysis buffer (20 mM $Na_2HPO_4$-$NaH_2PO_4$ pH 7.4, 50 mM NaCl, and 5% glycerol). Dialysis was performed at 4 °C with constant mixing using a magnetic stirrer. The dialyzed fractions were evaluated on a 12% SDS-PAGE.

## Biochemical characterization of *S. scabrisporus* L-asparaginase

To characterize the purified enzyme from *S. scabrisporus*, the asparaginase activity assay was performed using Nessler's reagent for the detection of ammonium [30].

For the standard assay, a time course of 0, 1, 3, 5, 7 and 10 minutes was established in which the amount of ammonium produced by asparaginase in 500 µL of reaction was quantified.

The conditions for the enzymatic reaction of L-asparaginase (500 µl) were: 200 µL of phosphate buffer (0.1 M $Na_2HPO_4$-$NaH_2PO_4$, pH 7.4) with 10 mM of substrate (L-asparagine) and 0.08 µg/mL of the purified enzyme (50 µl). The reaction was carried out at 37 °C. The reaction was stopped at the specified times by adding 250 µL of 1.5 M trichloroacetic acid. The quantification of the released ammonium was performed by measuring the absorbance at 420 nm in 96-well microplates using 25 µL of the enzymatic reaction products in 175 µL of water. 25 µL of Nessler reagent was added to the mixture and incubated for 10 minutes before measuring the absorbance at 420 nm. Additionally, tests were performed to determine the optimum pH and stability of the enzyme at temperatures of 37 °C and 4 °C. To determine the optimum pH of the enzyme, enzymatic activity tests were performed in the following buffers: Acetate (pH 3–5), Sodium phosphate (pH 6–8) and sodium carbonate (pH 9–11), all at a concentration of 0.1 M. To evaluate the stability of the enzyme, it was incubated for 17, 24 and 48 hours at 37 °C and the enzymatic activity was evaluated. The enzyme was also stored at 4 °C for 6 weeks and the enzymatic activity was subsequently measured. All enzyme activities were performed at least in triplicate. The averages of the activities obtained, along with their standard deviations, are shown.

## Comparison of the enzymatic activity of asparaginases from *S. scabrisporus* and *R. etli* under physiological conditions

To compare the enzymatic activities of the *S. scabrisporus* and *R. etli* asparaginases, the enzymatic kinetics of both were carried out simultaneously, as described above using 10 mM of L-asparagine as a substrate in phosphate buffer (0.1 M $Na_2HPO_4$-$NaH_2PO_4$, pH 7.4). The enzymatic activities were performed in triplicates.

## Kinetic characterization of asparaginases

To determine the kinetic parameters of the enzymes, enzymatic reactions were carried out in triplicate in time courses of 0, 1, 3, 5, 7, and 10 minutes; the substrate concentrations evaluated were from 0.5 mM to 20 mM. The nonlinear Michaelis-Menten model and the Origin and GraphPad programs were used to analyze the data.

## Cell culture and cytotoxicity assays

The T-cell lymphoblastic human leukemia cell line MOLT-4 (CRL-1582, ATCC) was cultured in the Roswell Park Memorial Institute medium (RPMI 1640), complemented with 2 mM L-glutamine, 10 mM HEPES, 1 mM sodium pyruvate, 4500 mg/L glucose, 1500 mg/L sodium bicarbonate, and 10% fetal bovine serum (v/v). Cells were incubated at 37 °C in 5% $CO_2$ in T-25 flasks. Cell proliferation was measured using the CellTiter 96® AQueous One Solution Cell Proliferation Assay kit (Promega, Madison, WI, USA, G3581), based on the [3-(4,5-dimethylthiazol-2-yl)-5-(3-carboxymethoxyphenyl)-2-(4-sulfophenyl)-2H-tetrazolium] (MTS) assay, following the manufacturer's instructions at different concentrations of the enzymes. Cell viability was determined after 24, 48, and 72 h of incubation with asparaginases from *S. scabrisporus* and *R. etli*, and as a positive control, Leunase (*E. coli* asparaginase). Cell viability without asparaginase was determined as a negative control. Cell viability assays were performed in triplicate, and the results shown are the average of three independent experiments.

**Statistical analysis.** Statistical analysis included average and standard deviation calculations, following by a two-way ANOVA with a Dunnett's post hoc test and $\alpha = 0.05$, by using GraphPad Software 9.0, San Diego, CA, USA.

## Acknowledgments

To Itandehui Betanzo and Oscar González-Davis for technical assistance.

## Author contributions

**Conceptualization:** Ricardo Rodríguez-Vargas, Francisca Villanueva-Flores, María Fernanda Gutiérrez-Chávez, Alejandro Huerta-Saquero.

**Formal analysis:** Ricardo Rodríguez-Vargas, Francisca Villanueva-Flores, Andrés Zárate-Romero, Alejandro Huerta-Saquero.

**Funding acquisition:** Alejandro Huerta-Saquero.

**Investigation:** Ricardo Rodríguez-Vargas, Francisca Villanueva-Flores, María Fernanda Gutiérrez-Chávez, Carlos Medrano-Villagómez, Andrés Zárate-Romero, Alejandro Huerta-Saquero.

**Methodology:** Ricardo Rodríguez-Vargas, María Fernanda Gutiérrez-Chávez, Carlos Medrano-Villagómez, Andrés Zárate-Romero, Alejandro Huerta-Saquero.

**Supervision:** Alejandro Huerta-Saquero.

**Writing – original draft:** Ricardo Rodríguez-Vargas, Alejandro Huerta-Saquero.

**Writing – review & editing:** Ricardo Rodríguez-Vargas, Francisca Villanueva-Flores, María Fernanda Gutiérrez-Chávez, Andrés Zárate-Romero, Alejandro Huerta-Saquero.

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
