## [Decision Letter · Decision Letter 0]

22 Sep 2025

Dear Dr. Huerta-Saquero,

Thank you for submitting your manuscript to PLOS ONE. After careful consideration, we feel that it has merit but does not fully meet PLOS ONE’s publication criteria as it currently stands. Therefore, we invite you to submit a revised version of the manuscript that addresses the points raised during the review process.

We look forward to receiving your revised manuscript.

Kind regards,

Jayaprakash Narayana Kolla

Academic Editor

PLOS ONE

**Journal Requirements:**

1. When submitting your revision, we need you to address these additional requirements. Please ensure that your manuscript meets PLOS ONE's style requirements, including those for file naming. The PLOS ONE style templates can be found at https://journals.plos.org/plosone/s/file?id=wjVg/PLOSOne_formatting_sample_main_body.pdf and https://journals.plos.org/plosone/s/file?id=ba62/PLOSOne_formatting_sample_title_authors_affiliations.pdf 2. We note that the grant information you provided in the ‘Funding Information’ and ‘Financial Disclosure’ sections do not match.  When you resubmit, please ensure that you provide the correct grant numbers for the awards you received for your study in the ‘Funding Information’ section. 3. If the reviewer comments include a recommendation to cite specific previously published works, please review and evaluate these publications to determine whether they are relevant and should be cited. There is no requirement to cite these works unless the editor has indicated otherwise. 

Reviewers' comments:

**Comments to the Author**

1. Is the manuscript technically sound, and do the data support the conclusions?

Reviewer #1: Yes

Reviewer #2: Yes

2. Has the statistical analysis been performed appropriately and rigorously?

Reviewer #1: Yes

Reviewer #2: No

3. Have the authors made all data underlying the findings in their manuscript fully available?

Reviewer #1: Yes

Reviewer #2: Yes

4. Is the manuscript presented in an intelligible fashion and written in standard English?

Reviewer #1: Yes

Reviewer #2: Yes

**Reviewer #1: ** Enzymatic properties of a L-asparaginase without secondary glutaminase activity from Streptomyces scabrisporus

Comments:

Overview:

The manuscript describes the effect of L-asparaginase from S. scabrisporus and R. etli in Acute Lymhocytic Leukemia. The author characterized L-asparaginase isolated from S. scabrisporus and R. etli . The enzymatic activity, stability, optimum pH and the kinetic parameters were evaluated. Further the cytotoxic effect of L-asparaginase was checked in in vitro model using MOLT-4 leukemic cell line. Overall, the study is well structured and planned. However, the author should revise the below mentioned minor points for publication.

Abstract:

Line no 6-7: The sentence is not clear, please rephrase the sentence “However, using L- asparaginase shows various obstacles due to immunogenic reactions ascribed to its bacterial origin”.

Line no 14: Please rephrase the sentence : “evaluate their viability as a potential chemotherapeutic treatment against ALL”

Results:

Line no 77-79: Rephrase the sentence, the sentence is not clear: “Likewise, the purification of L- asparaginase from R. etli followed the previously described procedure to purify the enzyme from S. scabrisporus”.

L-asparaginase was purified from S.scabrisporus using the standardized method described earlier for L-asparaginase purification from R. etli (ref)

Line no 82: Please elaborate the figure legend. What percentage of gel used, and what staining was used? Please add the necessary details in the figure legend.

How do you confirm the band at 34kDa is asparaginase? Yet it showed the enzymatic activity, it would be better if it could be identified by mass spec.

Line no 97-99: Rephrase the figure legend. Please change the figure 2 legend style as written in figure 3. Please check any PlosOne publication for figure legend format

Line no 112: Please mention the statistical test used

Line no 159-162: Though S.scabrisporus Asparaginase has low Kcat value than R.etli, the viability of MOLT-4 at later time points is comparatively better for Asparaginase from S.scabrisporus than R.etli. How do you explain this?

Methods:

Line no:329-330: Rephrase the sentence

**Reviewer #2: ** This manuscript presents original research on the purification, biochemical characterization, and kinetic analysis of L-asparaginases from Streptomyces scabrisporus and Rhizobium etli, with evaluation of their cytotoxic activity against MOLT-4 leukemic cells. The study addresses an important biomedical problem: identifying L-asparaginase variants with reduced immunogenicity and no glutaminase activity, which could improve therapeutic options for acute lymphoblastic leukemia (ALL).

Experimental Rigor: The cloning, expression, and purification protocols are well-detailed, with the use of HisTrap affinity chromatography and verification by SDS-PAGE appropriate. Enzymatic assays include replication and controls, using commercial E. coli asparaginase (Leunase) as a reference, strengthening the comparative analysis. Cell viability assays also incorporate proper negative and positive controls.

Kinetic Analysis: Michaelis-Menten kinetics are applied, and Km, Vmax, and kcat are calculated under both optimal (pH 10) and physiological (pH 7, 37 °C) conditions. A critical caveat is the use of very high substrate concentrations (up to 60 mM asparagine) to achieve saturation under physiological pH, far exceeding plasma levels (~50–100 µM). This can artificially inflate apparent Km values and complicate extrapolation to in vivo relevance. While the data demonstrate that both enzymes reduce leukemic cell viability in a time- and concentration-dependent manner, the eightfold increase in Km under physiological conditions suggests reduced substrate affinity that may limit therapeutic efficiency in vivo. This nuance is not fully explored in the conclusions.

Data Support and Statistical Rigor: The claim that S. scabrisporus L-asparaginase is glutaminase-free is experimentally supported, representing a significant advantage. However, while the manuscript reports mean ± SD and replicates, discussion of statistical significance for differences between enzymes or conditions is limited. Including appropriate statistical tests would strengthen the conclusions drawn from cell viability assays and kinetic comparisons. Replicates are indicated, but details about sample sizes and independent experiments are limited. For a rigorous evaluation, the inclusion of proper statistical tests with p-values or confidence intervals is recommended.

Overall Assessment: The manuscript is technically competent, scientifically valid, and presents original research with data that generally support its conclusions. Minor improvements in statistical rigor, data transparency, and text clarity are recommended before publication. Claims regarding potential clinical efficacy of S. scabrisporus asparaginase should be tempered given the discrepancy between assay substrate concentrations and physiological conditions.

**Do you want your identity to be public for this peer review?** For information about this choice, including consent withdrawal, please see our Privacy Policy

Reviewer #1: No

Reviewer #2: No

---

## [Author Response · Author response to Decision Letter 1]

1 Oct 2025

PONE-D-25-43083

Enzymatic properties of a L-asparaginase without secondary glutaminase activity from Streptomyces scabrisporus

PLOS ONE

RESPONSE TO REVIEWERS

Reviewer #1: Enzymatic properties of a L-asparaginase without secondary glutaminase activity from Streptomyces scabrisporus

Comments:

Overview:

The manuscript describes the effect of L-asparaginase from S. scabrisporus and R. etli in Acute Lymphocytic Leukemia. The author characterized L-asparaginase isolated from S. scabrisporus and R. etli . The enzymatic activity, stability, optimum pH and the kinetic parameters were evaluated. Further the cytotoxic effect of L-asparaginase was checked in in vitro model using MOLT-4 leukemic cell line. Overall, the study is well structured and planned. However, the author should revise the below mentioned minor points for publication.

RESPONSE: Thank you for your positive comments.

Abstract:

Line no 6-7: The sentence is not clear, please rephrase the sentence “However, using L- asparaginase shows various obstacles due to immunogenic reactions ascribed to its bacterial origin”.

RESPONSE: The sentence was modified as follows:

“However, due to the bacterial origin of L-asparaginase, it causes immunogenic reactions, and the cross-glutaminase activity that the enzyme exhibits cause ammonium accumulation and toxicity in different organs and tissues”.

Line no 14: Please rephrase the sentence: “evaluate their viability as a potential chemotherapeutic treatment against ALL”

RESPONSE: Thank you. The sentence was modified as follows:

“In this work, the L-asparaginases from S. scabrisporus and R. etli were purified and characterized, and the kinetic parameters of the enzymes were compared under physiological conditions”.

Results:

Line no 77-79: Rephrase the sentence, the sentence is not clear: “Likewise, the purification of L- asparaginase from R. etli followed the previously described procedure to purify the enzyme from S. scabrisporus”.

L-asparaginase was purified from S.scabrisporus using the standardized method described earlier for L-asparaginase purification from R. etli (ref)

RESPONSE: Thank you. For clarity, the sentence was modified as follows:

“The L-asparaginases from S. scabrisporus and R. etli were purified to near homogeneity by affinity chromatography using a HisTrap column (Cytiva). Electrophoretic analysis of the purified L-asparaginase from S. scabrisporus showed a 34 kDa band, consistent with the molecular weight of the recombinant enzyme. A protein yield of 2 mg/L was obtained. On the other hand, the purified L-asparaginase from R. etli showed a 38 KDa band, according to the estimated molecular weight of the recombinant enzyme. A protein yield of 3 mg/L was obtained (Fig. 1) [17].

Line no 82: Please elaborate the figure legend. What percentage of gel used, and what staining was used? Please add the necessary details in the figure legend.

How do you confirm the band at 34kDa is asparaginase? Yet it showed the enzymatic activity, it would be better if it could be identified by mass spec.

RESPONSE: We agree with the suggestion. Figure legend was modified as follows:

“Figure 1. Purified recombinant L-asparaginases from S. scabrisporus and R. etli. 12% SDS-PAGE: M.- Molecular weight markers (in kDa); lane 1) S. scabrisporus L-asparaginase (34 kDa); lane 2) R. etli L-asparaginase (38 kDa). Protein bands were stained with Coomasie Brilliant Blue R-250.”

Regarding the identification of recombinant asparaginases by mass spectrometry, we consider it unnecessary since the genetic constructs were verified by sequencing. The induction patterns of the recombinant proteins were observed by SDS-PAGE gels, where the corresponding bands are observed only in the induced samples, which, together with what the reviewer refers to regarding the enzymatic activity of asparaginase in the samples purified by affinity chromatography, we consider to be sufficient evidence to ensure that the asparaginases are really those we propose.

Line no 97-99: Rephrase the figure legend. Please change the figure 2 legend style as written in figure 3. Please check any PlosOne publication for figure legend format

RESPONSE: Figure legend was modified as follows:

“Fig 2. pH optimum and kinetic parameters of S. scabrisporus asparaginase. Enzymatic activity was determined at different pH (3 to 11) (A) Optimum pH was determined as 10. (B) At this pH, kinetic parameters were calculated and adjusted to the Michaelis-Menten model, shown in the inset”.

Line no 112: Please mention the statistical test used

RESPONSE: Done:

“Statistical analysis was performed using a two-way ANOVA with a Dunnett´s post hoc test and α = 0.05.”

Line no 159-162: Though S. scabrisporus Asparaginase has low Kcat value than R.etli, the viability of MOLT-4 at later time points is comparatively better for Asparaginase from S.scabrisporus than R.etli. How do you explain this?

RESPONSE: Excellent observation. Indeed, based on the data obtained regarding the lower catalytic capacity of S. scabrisporus asparaginase compared to R. etli asparaginase, it was expected that cell viability studies would result in greater inhibition activity with R. etli asparaginase, which occurred during the first 48 hours. However, after 72 hours, it was observed that S. scabrisporus asparaginase reduced cell viability more efficiently compared to R. etli asparaginase. The high stability of S. scabrisporus asparaginase can explain this result, suggesting that, despite its lower catalytic capacity, it persists for a longer time, reducing the availability of asparagine in the medium and, consequently, reducing cell viability at longer incubation times.

We included this new paragraph in the Discussion section:

“The high stability of S. scabrisporus asparaginase can explain this result, suggesting that, despite its lower catalytic capacity, it persists for a longer time, reducing the availability of asparagine in the medium and, consequently, reducing cell viability at longer incubation times”.

Methods:

Line no:329-330: Rephrase the sentence

RESPONSE: Done:

“Statistical analysis. Statistical analysis included average and standard deviation calculations, following by a two-way ANOVA with a Dunnett´s post hoc test and α = 0.05, by using GraphPad Software 9.0, San Diego, CA, USA.”

Reviewer #2: This manuscript presents original research on the purification, biochemical characterization, and kinetic analysis of L-asparaginases from Streptomyces scabrisporus and Rhizobium etli, with evaluation of their cytotoxic activity against MOLT-4 leukemic cells. The study addresses an important biomedical problem: identifying L-asparaginase variants with reduced immunogenicity and no glutaminase activity, which could improve therapeutic options for acute lymphoblastic leukemia (ALL).

RESPONSE: Thank you for your positive opinion of our work.

Experimental Rigor: The cloning, expression, and purification protocols are well-detailed, with the use of HisTrap affinity chromatography and verification by SDS-PAGE appropriate. Enzymatic assays include replication and controls, using commercial E. coli asparaginase (Leunase) as a reference, strengthening the comparative analysis. Cell viability assays also incorporate proper negative and positive controls.

RESPONSE: Thank you, we agree.

Kinetic Analysis: Michaelis-Menten kinetics are applied, and Km, Vmax, and kcat are calculated under both optimal (pH 10) and physiological (pH 7, 37 °C) conditions. A critical caveat is the use of very high substrate concentrations (up to 60 mM asparagine) to achieve saturation under physiological pH, far exceeding plasma levels (~50–100 µM). This can artificially inflate apparent Km values and complicate extrapolation to in vivo relevance. While the data demonstrate that both enzymes reduce leukemic cell viability in a time- and concentration-dependent manner, the eightfold increase in Km under physiological conditions suggests reduced substrate affinity that may limit therapeutic efficiency in vivo. This nuance is not fully explored in the conclusions.

RESPONSE: Agree. The conclusions were re-written as follows:

“The L-asparaginases from S. scabrisporus and R. etli were purified and characterized; both enzymes reduced the viability of MOLT-4 leukemic cells in a time- and concentration-dependent manner. Despite the low affinity of S. scabrisporus asparaginase for its substrate, asparagine, under physiological conditions, its stability and efficient glutaminase-free asparaginase activity highlight its potential as a viable alternative for the treatment of ALL”.

Data Support and Statistical Rigor: The claim that S. scabrisporus L-asparaginase is glutaminase-free is experimentally supported, representing a significant advantage. However, while the manuscript reports mean ± SD and replicates, discussion of statistical significance for differences between enzymes or conditions is limited. Including appropriate statistical tests would strengthen the conclusions drawn from cell viability assays and kinetic comparisons. Replicates are indicated, but details about sample sizes and independent experiments are limited. For a rigorous evaluation, the inclusion of proper statistical tests with p-values or confidence intervals is recommended.

RESPONSE: Agree. Replicates, statistical test used and more important experimental details were added in Material and Methods section and indicated in the figure legends.

Overall Assessment: The manuscript is technically competent, scientifically valid, and presents original research with data that generally support its conclusions. Minor improvements in statistical rigor, data transparency, and text clarity are recommended before publication. Claims regarding potential clinical efficacy of S. scabrisporus asparaginase should be tempered given the discrepancy between assay substrate concentrations and physiological conditions.

RESPONSE: Thank you. We agree on the comments.

The changes suggested by the reviewers significantly improve the manuscript, addressing the concerns of the reviewer. As suggested by the reviewer, we qualify the conclusions regarding the possible use of asparaginase in the clinic.

---

## [Editor Report · Decision Letter 1]

26 Oct 2025

Enzymatic properties of a L-asparaginase without secondary glutaminase activity from Streptomyces scabrisporus

PONE-D-25-43083R1

Dear Author,

We’re pleased to inform you that your manuscript has been judged scientifically suitable for publication and will be formally accepted for publication once it meets all outstanding technical requirements.

Kind regards,

Jayaprakash Narayana Kolla

Academic Editor

PLOS ONE

---

## [Editor Report · Acceptance letter]

PONE-D-25-43083R1

PLOS ONE

Dear Dr. Huerta-Saquero,

I'm pleased to inform you that your manuscript has been deemed suitable for publication in PLOS ONE. Congratulations! Your manuscript is now being handed over to our production team.

Kind regards,

on behalf of

Dr. Jayaprakash Narayana Kolla

Academic Editor

PLOS ONE